# Designing Biomimetic Conductive Gelatin-Chitosan–Carbon Black Nanocomposite Hydrogels for Tissue Engineering

**DOI:** 10.3390/biomimetics8060473

**Published:** 2023-10-03

**Authors:** Kamol Dey, Emanuel Sandrini, Anna Gobetti, Giorgio Ramorino, Nicola Francesco Lopomo, Sarah Tonello, Emilio Sardini, Luciana Sartore

**Affiliations:** 1Bio-Nanomaterials and Tissue Engineering Laboratory (BNTELab), Department of Applied Chemistry and Chemical Engineering, Faculty of Science, University of Chittagong, Chittagong 4331, Bangladesh; 2Department of Mechanical and Industrial Engineering, Materials Science and Technology Laboratory, University of Brescia, Via Branze 38, 25123 Brescia, Italy; e.sandrini005@studenti.unibs.it (E.S.); a.gobetti@unibs.it (A.G.); giorgio.ramorino@unibs.it (G.R.); luciana.sartore@unibs.it (L.S.); 3Department of Information Engineering, University of Brescia, Via Branze 38, 25123 Brescia, Italy; nicola.lopomo@unibs.it (N.F.L.); emilio.sardini@unibs.it (E.S.); 4Department of Information Engineering, University of Padova, 35131 Padua, Italy; sarah.tonello@unipd.it

**Keywords:** hydrogel, gelatin, chitosan, conductive carbon black, nanocomposite, cyclic compression, dissipation energy, anisotropy, tissue engineering

## Abstract

Conductive nanocomposites play a significant role in tissue engineering by providing a platform to support cell growth, tissue regeneration, and electrical stimulation. In the present study, a set of electroconductive nanocomposite hydrogels based on gelatin (G), chitosan (CH), and conductive carbon black (CB) was synthesized with the aim of developing novel biomaterials for tissue regeneration application. The incorporation of conductive carbon black (10, 15 and 20 wt.%) significantly improved electrical conductivity and enhanced mechanical properties with the increased CB content. We employed an oversimplified unidirectional freezing technique to impart anisotropic morphology with interconnected porous architecture. An investigation into whether any anisotropic morphology affects the mechanical properties of hydrogel was conducted by performing compression and cyclic compression tests in each direction parallel and perpendicular to macroporous channels. Interestingly, the nanocomposite with 10% CB produced both anisotropic morphology and mechanical properties, whereas anisotropic pore morphology diminished at higher CB concentrations (15 and 20%), imparting a denser texture. Collectively, the nanocomposite hydrogels showed great structural stability as well as good mechanical stability and reversibility. Under repeated compressive cyclic at 50% deformation, the nanocomposite hydrogels showed preconditioning, characteristic hysteresis, nonlinear elasticity, and toughness. Overall, the collective mechanical behavior resembled the mechanics of soft tissues. The electrical impedance associated with the hydrogels was studied in terms of the magnitude and phase angle in dry and wet conditions. The electrical properties of the nanocomposite hydrogels conducted in wet conditions, which is more physiologically relevant, showed a decreasing magnitude with increased CB concentrations, with a resistive-like behavior in the range 1 kHz–1 MHz and a capacitive-like behavior for frequencies <1 kHz and >1 MHz. Overall, the impedance of the nanocomposite hydrogels decreased with increased CB concentrations. Together, these nanocomposite hydrogels are compositionally, morphologically, mechanically, and electrically similar to native ECMs of many tissues. These gelatin-chitosan–carbon black nanocomposite hydrogels show great promise for use as conducting substrates for the growth of electro-responsive cells in tissue engineering.

## 1. Introduction

The natural extracellular matrix (ECM) consists of dynamic, highly complex, and hierarchically organized nanocomposites that govern and regulate cells’ fate and functions [1]. Moreover, the composition, density, nanostructure, and microstructure of the ECM quite differently vary between different tissue types [2]. As such, adopting a nanotechnological strategy for designing advanced nanocomposites to both better architecturally and functionally emulate the ECM, has gained intense interest [3,4]. Tissue engineering has emerged as a promising field for the regeneration and repair of damaged tissues and organs by copying the architectural and functional features of the native ECM. However, traditional tissue engineering approaches have limitations in mimicking the complex electrical properties of native ECM and tissues. Conductive nanocomposites offer an innovative solution to this challenge by combining conductive nanoparticles with biocompatible matrices to create a conducive microenvironment for cell growth and tissue regeneration [5,6]. The conductive properties of these nanocomposites facilitate better cell adhesion to the scaffold material [5]. Conductive nanocomposites can be used to deliver electrical stimulation to cells and tissues, which can promote cell proliferation, migration, and differentiation [5,7]. Electrical stimulation has been shown to accelerate tissue regeneration and improve the functionality of engineered tissues [6].

Therefore, conductive nanocomposites have emerged as a promising class of materials for tissue engineering, particularly for neural tissue engineering [8,9,10], cardiac tissue engineering [6,7,11,12], bone tissue engineering [13,14,15], cartilage tissue engineering [16,17,18], muscle tissue engineering [19,20], and skin tissue engineering [21,22,23]. They can be used to create neural scaffolds that support the growth and differentiation of neurons [10]. The electrical stimulation provided by the nanocomposite can also facilitate neural network formation and functional integration with host tissues [11]. In cardiac tissue engineering, conductive nanocomposites can be used to create scaffolds for heart muscle regeneration [11]. Electrical stimulation can help promote cardiomyocyte alignment and contractile behavior, leading to improved tissue functionality [12]. In bone tissue engineering, conductive nanocomposites can be integrated into scaffolds to support the growth and differentiation of bone-forming cells (osteoblasts) [13]. Electrical stimulation can also promote bone mineralization and enhance the overall healing process [14]. In addition to scaffolds, conductive nanocomposites can also be used as bioelectrodes for electrical stimulation applications as well as biosensors [24,25,26,27].

Additionally, the mechanical properties and degradation rates of the nanocomposites should be tailored to match the specific tissue engineering application [1,5,28]. It has been shown that conducting polymers, such as polyaniline and polypyrrole, can stimulate the attachment and proliferation of a variety of mammalian cell lines, including myoblasts, fibroblasts, and endothelial cells; however, these conducting polymers fail to mimic the physiological moduli of the ECM of various tissues [29]. This is why the incorporation of conductive nanomaterials, including metallic, inorganic, organic, or polymeric nanomaterials, into a suitable biopolymer matrix that can mimic a wide range of native ECM properties has become an alternative strategy to create multifunctional scaffolds that provide the appropriate structural, mechanical, electrical, and biological properties to foster healthy cell function and tissue formation [30,31].

Notably, there is a wide variety of carbon-based nanomaterials, such as carbon nanotubes, nanodiamonds, graphene oxide, and reduced graphene oxide, which can be used to develop electroconductive nanocomposite hydrogels [32,33,34]. However, compared to the abovementioned carbon-based nanomaterials, nanosized carbon black (CB) is an ill-explored nanofiller for biomedical applications [35]. As a sensing conductive filler, CB has a low cost, low density, and excellent intrinsic conductivity [36]. A CB-filled polymer composite has many advantages, including easy fabrication and superior environmental stability [36]. The electrical conductivity of CB is influenced by its particle size, aggregate shape and structure, porosity, and surface chemistry. The electrical conductivity of carbon black/polymer mixtures also depends on polymer characteristics such as the chemical structure, porosity, and processing conditions. The improved conductivity of polymer/carbon black mixtures is achieved by using CB of smaller particle size (larger surface area), lower particle density (higher particle porosity), and higher structure (better aggregation).

Here, we present a series of nanocomposite hydrogels, which are composed of CB nanoparticles homogeneously dispersed into a gelatin-based hydrogel system. In tissue engineering, gelatin (G), a denatured collagen product, is widely used because the material is biocompatible, biodegradable, inexpensive, and easy to use [37]. However, G alone is structurally and mechanically weak and dissolves in the physiological environment, which limits its application as a scaffold material in tissue engineering [37,38]. To overcome these limitations, a mild crosslinking strategy is applied where G acts as a backbone polymer and poly(ethylene glycol)diglycidyl ether (PEGDGE) acts as the crosslinking agent. By utilizing end epoxide groups, PEGDGE can react with multiple functional groups of G and act as a spacer between two natural macromolecules, reducing steric hindrance and facilitating cell adhesion [39]. Chitosan (CH), a partially deacetylated derivative of chitin, is used to further enhance the biomimetic properties of the hydrogel [40,41]. Most importantly, in the context of tissue engineering, CH is structurally similar to the native ECM, which facilitates cell-chitosan interaction [42,43,44]. Furthermore, CH possesses excellent biological properties such as biodegradability, anti-bacterial activity, and biocompatibility [42,44]. Moreover, many biological tissues including skeletal muscle, bone, cartilage, and the heart exhibit orientation-dependent anisotropic structures, and the structural, mechanical, and functional anisotropies of these tissues are critical for maintaining healthy physiological activities [45,46,47]. In this regard, anisotropic hydrogel scaffolds with geometrical resemblance to the ECM of anisotropic tissues are promising solutions for restoring their structural and functional integrity [48,49,50]. To achieve structurally and mechanically anisotropic porous hydrogel, we employed a unidirectional freezing technique using liquid nitrogen. We forced the degree of crosslinking through the concerted effect of chemical reactions in the solution and post-curing treatment. Together, the preparation of electroconductive and structurally stable macroporous nanocomposite hydrogels using gelatin, chitosan, and conductive carbon black was performed in this study in four consecutive steps: liquid-phase pre-crosslinking/grafting, oversimplified unidirectional freezing, lyophilization, and post-curing—this is the first report of this kind.

The overall aim of this study is to develop gelatin-based conductive hydrogel scaffolds via a mild processing condition, which includes aqueous media, various polymer assembly, and crosslinking chemistry facilitating gelation with CB nanomaterial, and to characterize the structural property, mechanical property, thermal property, morphology, and electrical conductivity of the synthesized nanocomposite hydrogels to find their potential applications as tissue engineering scaffolds.

## 2. Materials and Methods

### 2.1. Materials

Type A gelatin (pharmaceutical grade, 280 bloom, viscosity 4.30 mPs), produced from pig skin, was purchased from ITALGELATINE, Santa Vittoria d’Alba (CN), Italy. CH (molecular weight between 50,000–190,000 Da and degree of deacetylation 75–85%) was obtained from Fluka, Milano, Italy. PEGDGE (molecular weight 526 Da) was supplied by Sigma-Aldrich Co Milano, Italy. Conductive nanosized carbon black (Printex XE2B) was purchased from Degussa Huls Chemicals S.p.A. with an average particle size of 30 nm and BET surface area of 1000 m^2^/g. Ethylene diamine (EDA) and acetic acid were provided by Fluka, Milano, Italy.

### 2.2. Methods

#### 2.2.1. Synthesis of G/PEG/CH (CB) Nanocomposite Hydrogels

The G/PEG/CH (CB) nanocomposite hydrogel was prepared in different CB compositions, specifically 10%, 15%, and 20% of CB, following a slightly modified previously mentioned method [37]. For preparing G/PEG/CH (CB) nanocomposites with 10% CB, an amount of 0.80 g pulverized CB was dissolved in 60 mL water and treated with magnetic stirring and ultrasonication until completely dispersed. Then, gelatin granules (5.90 g) were added into the CB solution and magnetically stirred for 2 h at 45 °C to homogeneously dissolve the gelatin. After that, PEGDGE (1.4 g) was introduced into the mixture followed by EDA (70 mg) under continuous magnetic stirring at 45 °C. The chitosan solution of 2 wt.% was prepared by dissolving 0.7g CH in 1% aqueous acetic acid solution (35 g) and was subsequently stirred overnight. Then, the previously prepared chitosan solution (33 g) was added into a gelatin-PEG-CB reaction mixture and continuously stirred for another 1 h (with 5 min sonication after an interval of 20 min magnetic stirring) under the same temperature to obtain the homogeneous reaction mixture. Finally, the reaction mixture was poured into a plastic box, cooled to room temperature to form a gel, frozen with liquid nitrogen, lyophilized, and post cured at 45 °C for 2 h. The same sequences were applied for fabricating G/PEG/CH (CB) nanocomposites with 15 and 20% CB with the required amount of CB for each formulation. Three different G/PEG/CH (CB) nanocomposites were prepared; the formulations are tabulated in Table 1.

#### 2.2.2. Density and Porosity Measurement

The ethanol displacement method was used to measure the apparent density and porosity of the dry gel by soaking the pre-massed sample in a defined ethanol volume. The sample was previously placed under vacuum to remove the entrapped air, and the mass of the dry gel (W) was measured. Later, the sample was immersed into a graded cylinder containing a known volume (V_1_) of ethanol, and the total volume (V_2_) of ethanol and gel was recorded. The gel was carefully removed from the ethanol 5 h later, and the residual volume (V_3_) of the ethanol was measured. Finally, the total volume of gel was calculated as V= V_2_ − V_3_.

The apparent density (*ρ*) of hydrogel was calculated using the following equation:*ρ* = W/(V_2_ − V_3_)

The porosity (*ϵ*) of hydrogel was measured using the following equation:*ϵ* = (V_1_ − V_3_) × 100/(V_2_ − V_3_) 

#### 2.2.3. Swelling Ratio (%) Measurement

The swelling ratio of the hydrogel was estimated by soaking the previously weighed dry sample in the distilled water in a thermostatic bath at a temperature of 37 °C. At predetermined time intervals, the sample was removed from the distilled water, the excess surface water was removed by gently pressing it with absorbent paper, and the weight was accurately measured using an electronic analytical balance. The percentage swelling ratio (%) was determined as:S_R_ (%) = (W_s_ − W_d_) × 100/W_d_
where W_s_ and W_d_ indicate the wet and initial dry weight of the sample, respectively. Each experiment was conducted using five samples, and the average value was exposed with standard deviation.

#### 2.2.4. Chemical Structure Analysis

The chemical characterization of nanocomposite hydrogels was performed using Fourier transform infrared (FTIR) spectroscopy on dry hydrogels using a Thermo Scientific, Waltham, MA, USA, Nicolet iS50 FTIR spectrophotometer equipped with a PIKE MIRacle attenuated total reflectance attachment and recorded over a range of 400 to 4000 cm^−1^ at a resolution of 4 cm^−1^.

#### 2.2.5. Thermogravimetric Analysis

The exact amounts of CB in the nanocomposite hydrogels were evaluated by thermogravimetric analysis (TGA) run by a TGA 500 equipped using the Hi-Res-Dynamic method with a Mettler TG50 microbalance heating a sample of about 5 mg into an alumina crucible from room temperature to 700 °C at 50 °C/min under nitrogen flow (40 mL/min).

#### 2.2.6. Morphology Analysis

The morphology of the nanocomposite hydrogels was analyzed using a stereomicroscope (LEICA DMS 300) with reflected light. The dried samples were cut in parallel (direction of ice crystal growth) and perpendicular directions to the macroporous channels, and the morphologies of the surfaces and textures were observed using the stereomicroscope.

#### 2.2.7. Compression and Cyclic Compression Tests

The mechanical properties of the nanocomposite hydrogels were measured using a universal testing system (INSTRON series 3366) equipped with a 50 N load cell, in unconfined compression mode between two impermeable parallel plates. Specimens were cut from the sample bars into cuboid-shaped samples using a precision rotary saw. Before the tests, the as-prepared samples were soaked in distilled water at 37 °C for 2 h. The actual height (H), width (W), and thickness (T) of the specimens were measured using an optical traveling microscope. Prior to initiating the compression test, a pre-load of 0.005 N was applied in order to reduce the influence of surface artifacts. For the compression test, cuboid-shaped samples were compressed at a strain rate of 10 mm/min up to a maximum of 50% strain of the original heights, either along the parallel or perpendicular directions to the macroporous channels. The cyclic consecutive loading-unloading test was continuously performed for 10 cycles, as long as no significant change in curve shape was observed. At least three specimens were tested for each direction. For unconfined compression testing, the load–displacement (F–x) data are converted to stress–strain (*σ* − *ε*) data through simple geometrical relationships. Engineering stress was calculated by dividing the recorded force by the initial cross-section area. The engineering strain under compression was defined as the change in height relative to the original height of the individual specimen. The initial elastic modulus (stiffness) was calculated from the first compression cycle and determined as the linear segment slope of the compressive stress-strain within the range of 5–10% strain. Compressive strength was defined as the maximum stress at 50% strain.

The successive cyclic compression experiment involved a loading-unloading pattern to reach different maximum strains (%) at each cycle. The specimen was first compressed (loaded) to a maximum strain of 20% and then relaxed (unloaded). Sequentially, the specimen was compressed to 30% maximum strain and relaxed again, repeating the operation increasing the maximum strain to 40%, 50%, and 60%. The energy absorbed by the nanocomposite hydrogels was calculated by the cyclic compression stress-deformation curves. The total energy applied to the hydrogel during the loading, defined as compression energy (kJ/m^3^), was derived from the area included by the loading curve and horizontal axis, while the energy released by the hydrogel during the unloading, defined as relaxation energy (kJ/m^3^), was the area bounded between the unloading curve and horizontal axis. The hysteresis loop area indicates the dissipated energy due to the viscous nature of the hydrogels, so the dissipation energy (kJ/m^3^) loss during the hysteresis cycle was calculated as the difference between the area under the load curve (compression energy) and the discharge (relaxation energy) curve from the stress-deformation curve. The percentage of dissipation energy (%) was determined by dividing the dissipation energy by the compression energy. The same computations were performed for the cyclic compression test with gradually increasing maximum compressive strain from 20% to 60%, calculating the dissipation energy (kJ/m^3^) and percentage dissipation energy (%) at each maximum strain.

#### 2.2.8. Electrical Impedance Measurement

Preliminary electrical impedance analysis was carried out considering an alternate current (AC) regimen in the frequency range of 10^2^–10^7^ Hz at room temperature by using a commercial impedance analyzer (HP4194A). Nanocomposite hydrogels filled with different weight percentages of carbon black nanoparticles were tested in both dry and hydrated states. In order to highlight the resistive and capacitive behavior, the complex impedance associated with the samples was studied in terms of the magnitude and phase angle. Tests were performed in triplicate.

## 3. Results

### 3.1. Preparation of G/PEG/CH (CB) Nanocomposite Hydrogels and Their Physical Properties

Conductive nanocomposites have emerged as promising materials in the field of tissue engineering due to their unique properties that can enhance tissue regeneration and repair. These materials combine conductive elements such as conductive nanoparticles or conductive polymers with biocompatible polymers, providing electrical conductivity while supporting cell growth and tissue integration. Motivated by this, we synthesized G/PEG/CH (CB) nanocomposites using a versatile approach consisting of liquid-phase pre-crosslinking/grafting, a unidirectional freezing process carried out by liquid nitrogen subsequent freeze-drying, and a post-curing process, as shown in Figure 1.

The composition of the G/PEG/CH (CB) nanocomposite hydrogels is presented in Table 1. We calculated the exact amount of CB present in the final nanocomposite hydrogels with the help of TGA analysis (Figure 2a) and the results are illustrated in Table 1. TGA was performed with a heating rate of 50 °C/min to 700 °C under a nitrogen atmosphere. Under this operating condition, pristine conductive carbon black showed no weight loss, indicating indifference to thermal decomposition. However, G/PEG/CH (control sample) hydrogel showed significant weight loss and retained 19.80% of its original weight under the same conditions. Figure 2a showed the increased percentage residue with the higher CB content in the sample, and we applied the ‘subtraction’ approach to determine the final CB content, assuming that the increased percentage residue was attributed to thermally resistant CB. We calculated the final CB content in the nanocomposite hydrogel by subtracting the individual residue content from the control residue. The homogeneous dispersion of CB into the polymeric network is important to obtain the most beneficial contribution of CB incorporation. On visual inspection during synthesis, we could not observe any agglomeration of CB into the polymeric system. Furthermore, to confirm the uniform dispersion, we carried out TGA analysis of the G/PEG/CH (CB-1) nanocomposite using different sections of the same sample; namely, the top, middle, and bottom parts. Figure 2b exhibits no difference in residue for all sections, confirming uniform CB dispersion into nanocomposite hydrogels.

### 3.2. Chemical Structure Characterization

The chemical structure (functional groups) of the nanocomposite hydrogels was analyzed by FTIR using the washed samples; Figure 3 displays the IR spectrum of the control and nanocomposite hydrogels. The physico-mechanical properties of the nanocomposite hydrogels depend on the molecular level interaction of G-CB, CH-CB and PEG-CB. Gelatin showed prominent amine peaks around 1540 cm^−1^ (due to -NH bending vibrations and C-N stretching vibrations) and 1650 cm^−1^ (due to C=O stretching vibrations) [29]. The representative intense peak presented around 1096 cm^−1^, which was assigned to C-O stretching (ether bond C-O-C), validated the presence of PEG in the final hydrogels. The addition of CB neither split nor shifted any peaks or did not induce new peaks, indicating no chemical interaction with polymers at the molecular level.

### 3.3. Apparent Density, Porosity, and Swelling Ratio

The apparent density and total average porosity of the nanocomposite hydrogels were investigated using the ethanol displacement method and the results are shown in Table 1. The apparent density of G/PEG/CH (CB) nanocomposite hydrogels showed higher values than that of the G/PEG/CH control sample. However, the different amount of CB in these hydrogels appeared to slightly influence the density among the nanocomposites, showing similar density values around 0.13 g/cm^3^. At the same time, the decrease in hydrogel porosity was observed with the increase in the nanofiller CB amount. The increase in the carbon black amount as filler actually decreased the porosity and favored the formation of a hydrogel with a wider distribution of micropores. The control G/PEG/CH hydrogel showed a percentage porosity of 77%, whereas the G/PEG/CH (CB) nanocomposite hydrogels showed an average percentage porosity of 62.66%. We also investigated the effect of CB incorporation on the swelling ratio of the nanocomposite hydrogels. As shown in Figure 4, the swelling ratio of nanocomposites did not seem to be significantly influenced by the addition of CB. All nanocomposite hydrogels showed a higher swelling ratio (~730%) suitable for tissue engineering applications. Using a freeze-drying process resulted in a net-like porous structure, which enhanced the hydrogel’s ability to absorb large amounts of water.

### 3.4. Morphological Evaluation

Scaffolds for tissue engineering applications should have an interconnected porous structure for cell infiltration and the effective transport of nutrients and oxygen. The freeze-drying technique produced a thin skin with a dense texture (Figure 5a,b). However, beneath the skin, the hydrogel was highly porous (Figure 5c,d). A closer examination revealed this skin was composed of micropores (inset). However, removing the skin disclosed interconnected macropores of thin walls, which were themselves composed of micropores (Figure 5c,d). Overall, the freeze-drying technique used in our protocol produced highly porous hydrogels below a thin skin with a dense texture.

We also investigated the effect of CB incorporation into the pore morphology, and we analyzed the pore morphology in the parallel (side view) and perpendicular (cross-section view) directions of the ice crystal growth, as shown in Figure 6a–f. With 10% CB content, the hydrogel showed an anisotropic pore morphology: larger macroporous channels and random pores formation along and normal to the ice crystal growth, respectively (Figure 6a,c). The unidirectional freezing process allowed water molecules to become preferentially crystallized (ice crystals) along the thermal gradient (bottom to top) induced by bathing in liquid nitrogen. The lyophilization process sublimated ice crystals leaving behind the interconnected macroporous channels along the thermal gradient (Figure 1a). Surprisingly, a higher CB content compromised the anisotropic nature of the hydrogels as well as the pore sizes. A cross-section view of the G/PEG/CH (CB-2) hydrogel revealed a denser texture with smaller-sized pores (Figure 6b). Smaller channels, both in terms of length and diameter, were observed with a 15% CB content along the direction of ice crystal growth (white color arrows indicate the smaller channels) (Figure 6e). A further increment of CB concentration to 20% (G/PEG/CH (CB-2)) imparted more compact structures in both directions, completely nullifying the anisotropic pore morphology and introducing an isotropic structure (Figure 6e). The most plausible explanation behind the transition of anisotropic to isotropic pore morphology with the increased CB content might be that the higher CB concentration restricted the directional ice crystal growth. Excitingly, this CB concentration-imposed pore morphology accordingly triggered a mechanical response (discussed below).

### 3.5. Compressive Mechanical Properties

To explore the effect of CB incorporation on the mechanical properties, a uniaxial unconfined compression test was carried out on swollen samples in distilled water at 50% strain (Figure 7). It is obvious from Figure 7 that the nanocomposite hydrogels showed reversible behavior (returned to their original position) during the compression test. The initial elastic modulus (stiffness) was calculated from the initial linear regions of the stress-strain curves (3 to 10% strain) and the stress at 50% strain was considered at strength. All the hydrogels were compressed in two directions (parallel and perpendicular) to examine the effect of the pore morphology on the mechanical stiffness and strength; the average values of the results are summarized in Table 2. Figure 8a–c shows the stress-strain curves of nanocomposite hydrogels obtained from mechanical data when samples were subjected to compression in both parallel and perpendicular directions to the ice crystal growth. It was noticed that the anisotropic mechanical phenomenon became negligible, increasing the CB amount in the nanocomposite hydrogels. Clearly, the G/PEG/CH (CB-1) nanocomposite hydrogel showed a significant anisotropic mechanical property, displaying an anisotropic ratio-ratio of the modulus in the parallel direction to the perpendicular direction-of 3.70 (Figure 8a). Notably, the anisotropic mechanical response significantly reduced with a 15% CB content (G/PEG/CH (CB-2), Figure 8b), while it completely diminished with a 20% CB content (G/PEG/CH (CB-3), Figure 8c). This behavior might be explained by the more compact structure due to the larger amount of conductive carbon black aggregated with increasing CB content in the gel structure, which did not allow water molecules to accrete in larger crystals during the freeze-casting process. This reduced the gel matrix voids and made the macroporous channels less aligned with the temperature gradient generated by nitrogen freezing, as shown in Figure 1b.

The pore morphology of the nanocomposite hydrogels imparted these direction-dependent mechanical responses. Notably, G/PEG/CH (CB-1) nanocomposite hydrogel showed around a four-fold (0.063 MPa to 0.230 MPa) increase in stiffness when compressed in the parallel direction to the macropore channels compared to the perpendicular direction (Table 2). The two best plausible explanations for the increased stiffness when compressed in the parallel direction in contrast to the perpendicular compression for G/PEG/CH (CB-1) nanocomposite hydrogel might be due to: (i) the large lamellae, which act as pillars reinforcing the hydrogel, and (ii) the presence of more entrapped pressurized water between two compressive plates so that water could not be easily squeezed out [37].

The direction-dependent stiffness of anisotropic hydrogel was correlated with their anisotropic pore structures. As a possible confirmation of this, we could notice that a greater extent of macroporous channels along the direction of freezing in the G/PEG/CH (CB-1) nanocomposite hydrogel made it an anisotropic hydrogel. On the other hand, a dimensional reduction of the walls of macrochannels in the hydrogel with 15% of CB corresponded to a loss of the anisotropic behavior. In the G/PEG/CH (CB-2) nanocomposite hydrogel, the difference between the directional-dependent stiffness and strength was so little that we could treat it as an intermediate hydrogel in terms of mechanical anisotropy. The typical channel-like morphology disappeared in the G/PEG/CH (CB-3) nanocomposite hydrogel, and the more compact pore structure probably allowed the water to squeeze out in a similar way in both directions, making it a perfectly isotropic hydrogel. Figure 9 demonstrates the trend of compressive stiffness and strength variation with CB content. Figure 9 and Table 2 show a significant improvement in stiffness and strength upon the addition of CB into the hydrogel system. For example, a three-fold increase in both stiffness and strength was observed with 20% CB content. However, the nanocomposite hydrogel containing 10% CB displayed similar stiffness and strength values of G/PEG/CH alone in both directions.

In addition to investigating the effect of CB content on the mechanical stability of the nanocomposite hydrogels, cyclic compression tests with ten loading-unloading cycles were also performed. Figure 10 shows the cyclic compressive stress-strain curves for up to 10 consecutive cycles at 50% maximum strain without waiting time in the parallel and perpendicular directions to the macroporous channels of the G/PEG/CH (CB-1) nanocomposite hydrogel in wet conditions. Figure 11 presents the cyclic compressive stress-strain curves of the G/PEG/CH (CB-2) and G/PEG/CH (CB-3) nanocomposite hydrogels, as well as the corresponding stress-time curves.

As displayed in Figure 10a, the stress-strain curves of the G/PEG/CH (CB-1) nanocomposite hydrogel clearly showed two different pathways during the loading and unloading cycles, resulting in hysteresis loops. Regarding the cyclic compression in a direction parallel to the macroporous channels, a pronounced deviation in the stress-strain loading-unloading curves was shown between the first and second cycles, which indicated the occurrence of irreversible damage during the first cycle. It has been hypothesized that those smaller lamellae bridges connecting larger lamellae might be fractured during the first cycle; furthermore, the characteristic plateau region was affected by these major micro fractures in the hydrogel, involving a more collapsed structure and reduced buckling resistance of the lamellae for the consecutive cycles [37,38]. However, a slighter reduction in the slope and maximal stress at 50% strain was observed at each compressive cycle from the second to fifth consecutive loading-unloading cycle in the material, due to the occurrence of other minor micro fractures. After the fifth cycle, all following hysteresis loops were closely overlapped, indicating that the hydrogel achieved a mechanically stable structure. It was noticed that all unloading curves returned to 0% strain, indicating full shape recovery of the hydrogels even at 50% deformation. The stress responses (Figure 10c) of the G/PEG/CH (CB-1) nanocomposite hydrogel compressed at a constant strain level of 50% were shown as a function of time during the 10 cycles. It was observed that the induced stress exhibited a transition phenomenon: during the first cycle, the induced stress was 0.064 MPa, but during subsequent cycles, it gradually decreased and became stationary, reaching a constant value of 0.056 MPa after multiple cycles. This mechanical response was the outcome of a stress-softening tendency, described as Mullins’ effect, which is characterized by a lower resulting stress for the same applied strain. It is often reported in filled and non-filled rubber-like materials, and such behavior is known as “preconditioning” [51].

Figure 10b shows the stress-strain curves resulting from the cyclic loading-unloading curves compressed in the perpendicular direction to the macroporous channels. In this case, the stress-strain curves indicated a very small variation between the first and second cycle curves, indicating some minor irreversible micro fractures during the first loading. However, the second cycle and all subsequent hysteresis loops were overlapped, suggesting that the hydrogel had a good reversible behavior. The corresponding stress-time plot (Figure 10d) also confirmed the mechanical stability. The maximal stress slightly reduced during the second cycle; however, it rapidly reached a stationary value after subsequent cycles. The mechanical responses observed in the G/PEG/CH (CB-2) and G/PEG/CH (CB-3) nanocomposite hydrogels during compression testing (Figure 11) were similar. During the first loading, some sort of micro fracture occurred, however, from the second cycle onward, and after subsequent cycles, nearly identical hysteresis loops were observed. They exhibited a preconditioning behavior up to the fifth cycle, as a result of the stress-softening effect explained in the previous paragraph.

Cyclic stress softening can be characterized by the amplitude of a normalized stress decrease and by the number of cycles needed to reach a stabilized state. This effect was also evaluated by calculating the ratio of the maximum stress of every cycle to the maximum stress of the first cycle (normalized stress). As observed in Figure 12, the normalized stress over 10 cycles preserved at least 85% of the maximal stress reached in the first cycle for G/PEG/CH (CB-1) in the parallel direction, G/PEG/CH (CB-2) and G/PEG/CH (CB-3). Lower stress-softening was observed particularly for G/PEG/CH (CB-1) in the perpendicular direction, showing an amplitude of a normalized stress decrease of 5%, confirming that discussed above.

In summary, during the first cycles, the hydrogels demonstrated the Mullins effect, with a reduction in stiffness and stress at every cycle. After a few cycles, the material behavior stabilized, and the hydrogels were able to sustain a compressive strain of 50% with full strain recovery. All the nanocomposite hydrogels maintained their original shape and their load-bearing capability up to a high level of deformation. It might be possible that many microscopic flaws were created within the hydrogels, but no macroscopic cracks propagated; as a result, the hydrogels were not fractured at a macroscopic level. Overall, the G/PEG/CH (CB-1) nanocomposite hydrogel exhibited structural and mechanical anisotropy, natural tissue-like preconditioning, characteristic hysteresis, nonlinear elasticity, and energy dissipation—making it a biomimetic anisotropic hydrogel more closely resembling the native ECM of many soft tissues.

Additionally, dissipation energy calculated for each cycle also showed sharply reduced absorbed energy after the first cycle, but this was nearly constant for subsequent cycles (Figure 13a). In the case of G/PEG/CH (CB-1) nanocomposite hydrogel, when compressed in parallel to the macroporous channels, the calculated dissipation energy was double compared to those in the perpendicular direction (Figure 13a). Furthermore, the slope of this decrease was more marked along the parallel direction before the tested samples reached a constant value after a few cycles for both compression directions. This might be explained by the more stable structure in the perpendicular direction, as previously depicted. Additionally, as shown in Figure 13b, the percentage of dissipated energy was the highest for the first cycle for all hydrogels. However, the percentage of dissipation energy was found to be similar after the fifth to the subsequent tenth cycle, further confirming the achieved mechanical stability after a few cycles. Excitingly, all nanocomposite hydrogels maintained their original shapes after undergoing such a high level of deformation.

Moreover, nanocomposite hydrogels were subjected to cyclic compression with increasing maximum strain ranging from 20 to 60%; the stress-strain curves are represented in Figure 14a–d. In addition, energy dissipation was calculated from the hysteresis loop with different strain levels (Figure 15). Figure 14 and Figure 15 highlight that hysteresis became apparent above 30% strain and sharply increased with increasing strain levels, indicating the increased absorption of energy during higher deformation. For all cases, the hysteresis loops area became larger with increasing maximum strain. Furthermore, as presented in Figure 15a, the G/PEG/CH (CB-1) nanocomposite hydrogel showed an exponentially increasing trend in dissipation energies with an increase in maximum strain, suggesting the capability of the hydrogel to effectively dissipate energy at larger deformations, and also indicating a gradual fracture process of the gel network while reaching higher deformation levels with respect to the previous cycle. The slightly higher dissipated energies for the parallel compression of G/PEG/CH (CB-1) with respect to the perpendicular compression might be due to the migration of more pressurized water throughout the porous network. The exponential growth of dissipated energies at higher deformation levels became more apparent when the CB amount was increased to 15% and 20%, achieving values of 12 kJ/m^3^ and 16 kJ/m^3^ for 60% strain, respectively, which were twice and three times more than the energy dissipated at 10% CB content.

Percentage dissipation energy is often used to quantify the energy dissipation ability of a tough hydrogel. A higher level of deformation involves more dissipation due to the increased stress applied to the material (higher friction). By applying a greater force, a higher loading energy was conferred to the material and the dissipation energy increased for the water movement. As shown in Figure 15b, the G/PEG/CH (CB-1) and G/PEG/CH (CB-2) nanocomposite hydrogels exhibited similar dissipated energy capacities, which slightly increased with higher deformations. At the same time, G/PEG/CH (CB-3) demonstrated lower dissipation ability, but still increased the strain up to 60%, suggesting a less pronounced bond rupture capacity, increased elastic properties, and more deformation tolerance. It is noteworthy that, with 20% CB, the percentage dissipation energy was more constant, and it was probably able to sustain higher deformation, which led to higher values of percentage dissipation energy (not necessarily linearly increasing over 60% deformation). This CB nanofiller reinforced the hydrogel by having a greater effect on strength than modulus. This involved high compression energy during loading; as the deformation applied to the material increased, the dissipation energy increased but the compression energy increased relatively more (due to the reinforcement effect). These results clearly suggested that the nanocomposite hydrogel possessed CB concentration-dependent and strain-dependent energy dissipation behavior, and effective energy dissipation occurred at the higher strain deformation.

It is generally accepted that the stiffness and toughness of conventional materials are two opposite mechanical parameters. Interestingly, our G/PEG/CH (CB) nanocomposite hydrogel could simultaneously increase both stiffness and toughness with increasing CB concentration. This could be attributed to the hybrid physical and chemical crosslinking in the nanocomposite hydrogel. The enhancement in the elastic modulus (an indicator of stiffness) of the G/PEG/CH(CB) nanocomposite hydrogels was attributed to the increase in the elastically effective G/PEG/CH chains and the reinforcing effect of the carbon black acting as a filler. The rupture (or peeling) of physically adsorbed G/PEG/CH chains from CB was accompanied by energy dissipation, which consequently improved the crack resistance of the hydrogels. In addition, as more G/PEG/CH chains were adsorbed on CB particles, more energy was dissipated. Therefore, the toughness of the G/PEG/CH (CB) nanocomposite hydrogels continued to increase with the CB concentration. Taken together, a slight decrease in the stress and no residuary strain during the consecutive loading-unloading cycles demonstrated the elasticity, excellent shape-recovery properties, and good mechanical stability of the nanocomposite hydrogels.

### 3.6. Electrical Impedance

To highlight the resistive and capacitive behaviors, the complex impedance associated with the samples was studied in terms of the magnitude and phase angle in dry and wet conditions, as presented in Figure 16 and Figure 17, respectively. Considering dry testing, the results reported good optimally tunable electrical properties with a characteristic frequency of transition between resistive- and capacitive-like behaviors, depending on the CB content, despite the presence of noise at low frequency. Such a result appears to be promising in view of the application of electrical stimuli, even when compared with more complex materials. The electrical properties in wet conditions, which were more physiologically relevant, showed low magnitude at high frequencies (>1 MHz) with a decreasing phase angle, confirming a capacitive behavior (Figure 17). At lower frequencies (<1 kHz), impedance decreased with increased CB content, reaching a steady-state value in a range between 1 kHz and 1 MHz, with a near to zero phase angle confirming a resistive behavior. Overall, the impedance of the nanocomposite hydrogels decreased with increased CB concentrations; furthermore, it is worth noting that the G/PEG/CH scaffold without CB presented, within the very same frequency range, an overall value of impedance magnitude of about 10 kΩ. Additionally, a change in electrical conductivity, with and without biological cells and along with various protocols, was in progress. Since these nanocomposite hydrogels were prepared with the aim of developing novel scaffolds for tissue engineering applications, it is of utmost importance to evaluate the biocompatibility of these types of hydrogels, which is presently ongoing. However, our previous studies on G/PEG/CH hydrogels using human mesenchymal stem cells (hMSCs), assessing their tissue regeneration capability (chondrogenic and osteogenic differentiation), confirmed that this hydrogel scaffold was biocompatible and suitable for cell growth, chondrogenic and osteogenic differentiation, and mineralization [42,52].

## 4. Conclusions

Conductive nanocomposites have shown immense potential in tissue engineering due to their ability to support cell growth, promote tissue regeneration, and provide electrical stimulation to engineered tissues. These results revealed that the developed synthesis method was amenable enough to accommodate nanomaterials into macroporous hydrogels under mild conditions. The incorporation of conductive carbon black (CB) (10, 15, and 20 wt.%) significantly improved electrical conductivity and enhanced mechanical properties with the increased CB content. Interestingly, nanocomposite hydrogel with 10% CB provided both anisotropic morphology and mechanical properties, whereas anisotropic pore morphology diminished at higher CB concentrations (15 and 20%), imparting a denser texture. In general, under repeated compressive cycles at 50% deformation, all nanocomposite hydrogels showed nonlinear elasticity, toughness, preconditioning, and characteristic hysteresis. Overall, the collective mechanical behavior resembled the mechanics of soft tissues. The electrical properties of the nanocomposite hydrogels conducted in wet conditions, which is more physiologically relevant, showed a decreasing magnitude with increased CB concentrations, with a resistive-like behavior in the range of 1 kHz–1 MHz and a capacitive-like behavior for frequencies <1 kHz and >1 MHz. Overall, the impedance of the nanocomposite hydrogels decreased with increased CB concentrations. This work is still in the development and optimization stages. Finally, the combination of tissue-like morphological and mechanical behaviors, along with good electrical conductivity, could lead to their use in various applications.

## Data Availability

The raw/processed data required to reproduce these findings cannot be shared at this time as the data also forms part of an ongoing study.

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
