# Peer review of "Designing Biomimetic Conductive Gelatin-Chitosan–Carbon Black Nanocomposite Hydrogels for Tissue Engineering"

_biomimetics, 2023, doi:10.3390/biomimetics8060473_

Round 1

Reviewer 1 Report

This manuscript by Dey et al. described a conductive hydrogel consisting of gelatin, chitosan, and carbon black for potential tissue engineering applications. The hydrogel was characterized in terms of its mechanical, physical, and electrical properties. Overall, the hydrogel represents a promising material to fabricate electro-responsive cells in tissue engineering. I have the following comments for revision.

1.     While the authors mentioned that evaluating the biocompatibility of the hydrogel will be their future work, I think the authors should at least demonstrate the in vitro biocompatibility of the hydrogel in this manuscript.

2.     In Figure 11, the authors should show results generated by more cycles of loading.

3.     The following recently published manuscripts are related to conductive hydrogel and application of conductive hydrogel for neural/muscle tissue engineering: doi.org/10.3389/fbioe.2022.912497; doi.org/10.1038/s41467-023-37948-1; doi.org/10.1002/EXP.20210035; doi.org/10.1093/rb/rbab035; https://doi.org/10.1002/EXP.20220006. The authors should consider discussing them in the manuscript. 

Author Response

Dear Reviewer, 

I would like to express my sincere gratitude for taking the time to review our manuscript. Your feedback and suggestions have been immensely valuable in enhancing the quality of our manuscript. We carefully considered all of your suggestions and made the recommended changes to improve the manuscript. 

Best regards,

Reviewer 2 Report

This manuscript reports the fabrication of conductive gelatin-chitosan–carbon black nanocomposite hydrogels, and the corresponding characterizations were well-conducted and discussed. The manuscript is well-organized and written. However, some issues require to be solved before the publication.

1.      The authors describe the prepared nanocomposite hydrogels are “biomimetic” in the title. But I do not understand how they are biomimetic. The hydrogel is anisotropic, porous, and conductive. Is the morphology and the functions of complex electrical property of native ECM and tissue is similar to the characteristic and function of the hydrogel? If it is not, the “biomimetic” is not suitable. If it is yes, more explanation is required.

2.      The experiment about the application of the prepared hydrogels in tissue engineering is not conducted. Therefore, the existence of “tissue engineering” in the title may easily mislead the readers. I suggest deleting the “tissue engineering” from the title.

3.      Explanation about the relationship between anisotropic pore morphology and the direction of ice growth is required. A more detailed schematic may be helpful.

4.      How to guarantee the anisotropic pore morphology is same for different samples? How to control the direction of ice growth in the experiment?

5.      Some spelling errors exist in the text, such as “ml”, and “oC”. The blank between number and “%” is not required. A blank is required between the number and “mm” for describing the scale bars in Figure 6.

6.      The unit in the vertical coordinate of two curves in Figure 2 is “%”. Therefore, the vertical coordinate value of the curve is incorrect.

7.       Some recent literatures should be cited, such as Exploration 2022, 2, 20210083; Exploration 2022, 2, 20210029; Chinese Chemical Letters 2022, 33, 871-876.

Author Response

(The authors gave the same response as above.)

Reviewer 3 Report

Mechanical properties of G/PEG/CH/CB composites have been studied. The subject is potentially interesting and publishable. Some good material characterization can be found in the manuscript. However, there are many points which should be addressed, clarified and/or discussed by the authors in the revised version. Therefore, I suggest major revision of the manuscript based on the following comments:

1. The abstract is mostly similar to an introduction part. The main results of this work should be further highlighted in abstract. Hence the abstract of the manuscript needs a significant revision.

2. Could the authors comment on the water content of the chitosan-based samples?

3. The authors have mentioned that “… adopting nanotechnological strategy for designing advanced nanocomposites to better emulate the ECM, both architecturally and functionally, has gained intense interests [3].”. But, ref. [3] is an old one, although it is an important reference. Therefore, this statement can be further support by some recent articles such as (Engineered biomimetic membranes for organ-on-a-chip) for biomimetic membranes and [DOI: 10.2174/1574888X18666221227142055] for specific bone scaffolds.

4. Updating of the references in the introduction section is highly necessary. Many of the key references are old ones.

5. It has been mentioned that “Conductive nanocomposites offer an innovative solution to this challenge by combining conductive nanoparticles with biocompatible matrices to create a conducive microenvironment for cell growth and tissue regeneration [4].”. However, ref. [4] is a little old and so this statement can be further supported by recent literatures such as (electrically conductive carbon-based (bio)-nanomaterials for cardiac tissue engineering).

6. In lines 55-57, each item can be supported just by one important reference. The other references (the second and/or third ones) can be removed.

7. The application of chitosan as antibacterial, flexible as well as biocompatible template for tissue engineering was previously reported in [J. Mater. Chem. B, 2018,6, 7427-7438] and [Applied Surface Science 301 (2014) 456-462]. These should be mentioned in the revised version. In addition, the novelty of this work should be further highlighted as compared to the previous ones.

8. The number of figures of the manuscript is very high. Some of the figures can be merged for reducing the number of figures.

9. It has been mentioned that “However, it’s essential to choose appropriate conductive nanomaterials and matrix materials that are biocompatible, non-toxic, and stable to create effective tissue engineering platforms”. But, this statement seems as a negative and irrelevant statement as compared to the last ones. So, I suggest removing it. The next statement (Additionally, the mechanical properties and degradation rates of …) is Ok, because it refers to the main subject of the manuscript.

10. The manuscript is more devoted to the characterization of mechanical properties of the samples rather than the tissue engineering. However, the title is concentration only on tissue engineering. In fact, the title seems a little misleading and so should be improved.

11. The authors have mentioned that “there is a wide variety of carbon-based nanomaterials, such as carbon nanotubes, nanodiamonds, graphene oxide, and reduced graphene oxide can be used to develop electroconductive nanocomposite hydrogels [23-26].”. But. Ref. [23] is not related to these types of carbon materials and should be removed. Ref. [26] is related to conductive nanomaterials rather than specifically to carbon ones and so should be removed. For nanodiamons the following review articles is suggested: (synthesis and characterization of nanodiamond reinforced chitosan for bone tissue engineering).

12. It has been written that “Chitosan (CH), a partially deacetylated derivative of chitin, is used to further enhance the biomimetic properties of the hydrogel.”. This needs to be supported by, e.g., [J Tissue Sci Eng 2017, 8:3, 1000212] and [Journal of Controlled Release 350 (2022) 175-192].

13. Could the authors comment on the surface charge of the samples?

14. The authors should give some evidence relating to the electrical conductivity of the samples, due to the presence of the CB.

Some revisions can be considered during the improvement of the manuscript. 

Author Response

(The authors gave the same response as above.)

Round 2

Reviewer 1 Report

The authors have addressed the comments satisfactorily.

Reviewer 3 Report

The manuscript has been revised based on the comments and now can be considered for publication.